# A Longitudinal Analysis of Equine Asthma Presentation and Response to Treatment Using Lung Function Testing and BAL Cytology Analysis in Combination with Owner Perception

**DOI:** 10.3390/ani13213387

**Published:** 2023-11-01

**Authors:** Tyler-Jane Robins, Daniela Bedenice, Melissa Mazan

**Affiliations:** Cummings School of Veterinary Medicine, Tufts University, North Grafton, MA 01536, USA

**Keywords:** BAL, lung function testing, equine, airway hyperreactivity, clinical score, heaves, equine asthma

## Abstract

**Simple Summary:**

Equine asthma (EA), a chronic non-infectious inflammatory disease of the lower airways, is a pervasive and important cause of poor respiratory health in horses. The clinical presentation ranges from mild decrements in peak performance to crippling respiratory impairment. Horses with severe equine asthma may be diagnosed and effectively treated using clinical scoring alone, but more subtle disease often requires lung function testing and cytologic evaluation of lower airway respiratory secretions (via bronchoalveolar lavage, BAL) in order to arrive at a correct diagnosis and allow for the best treatment. Published data are lacking to describe the longitudinal course of disease, as characterized by a pulmonary specialty clinic along with owner descriptions. This study examines horses with at least two visits to a specialty pulmonary clinic for evaluation of EA, describes and quantifies the types and range of clinical presentations that fall under the umbrella term of EA in horses in New England, considers the reliability of owner perception of horses clinical signs through a newly developed owner complaint score, and documents and assesses the diagnostic methods used at the Tufts Cummings School of Veterinary Medicine, as well as the response of horses with EA to treatment and time.

**Abstract:**

(1) Background: Equine asthma (EA) is a pervasive and important cause of poor performance and respiratory morbidity in horses. Diagnosis of EA includes an owner complaint, clinical scoring, lung function testing, and cytological analysis of bronchoalveolar lavage (BAL) cytology. There is a paucity of information about the longitudinal course of the disease using these outcome assessments; thus, this study sought to describe and quantify, in horses with more than one visit to a specialty pulmonary clinic in New England, the type and range of clinical presentations with an eventual diagnosis of EA. It also aimed to develop and compare the outcomes of scoring systems for owner complaints and veterinary assessments, document and assess the diagnostic methods used, and evaluate the response of the horses to treatment and time. (2) Methods: This study was a retrospective, cross-sectional, STROBE-compliant observational analysis of equine patients who visited the Tufts Cummings Hospital for Large Animals (HLA) for evaluation of equine asthma (EA) from 1999–2023. The horses were categorized as having mild–moderate (mEA) or severe EA (sEA) using the ACVIM consensus statement guidelines. After excluding those with inadequate documentation or only one visit (*n* = 936), a total of 76 horses were included in the study. Of the 197 visits, 138 (70.0%) resulted in a diagnosis of mEA and 45 (22.8%) resulted in a diagnosis of sEA. Demographic information, owner complaints, clinical examination and scoring, lung function testing, BAL cytology, and recommendations for environmental remediation and pharmacologic treatment were recorded for all the visits. The data were analyzed for agreement between owner complaints (complaint score, CS) and clinical examination findings (examination score, ES), changes in CS and ES, lung function testing, and BAL cytology over time, with 197 visits recorded. (3) Results: A comparison between the CS and ES showed that the owners were more likely than veterinarians to detect cough, and a decrease in cough was the most common owner observation after treatment. The response to the histamine challenge, used to detect airway hyperreactivity, was significantly improved with treatment or time in the horses with mEA, whereas baseline lung function did not significantly change in mEA or sEA. (4) Conclusions: Owners can be astute observers of clinical signs, especially cough, in EA. Tests of airway hyperreactivity are more successful in detecting changes in mEA than are baseline lung function testing and assessment of BAL cytology.

## 1. Introduction

Equine asthma (EA), an umbrella term for chronic, non-infectious inflammation of the lower airways, is second only to musculoskeletal disease as a cause of poor performance in the equine industry, resulting in lost training days, failure to compete, and high veterinary costs. In mild equine asthma, previously known as inflammatory airway disease (IAD), small airway disease, or mild bronchiolitis, the clinical presentation is often subtle and even indiscernible unless the horse is exercising at maximum speed. The typical clinical signs in mild/moderate EA (mEA/IAD) are poor performance with intermittent coughing, but normal respiratory rate and effort are present at rest [1]. Severe EA (sEA), previously known as heaves or recurrent airway obstruction (RAO), on the other hand, is characterized by obvious clinical signs including frequent coughing, visibly increased resting respiratory effort, and severe exercise intolerance [2].

Horses with EA often present with varying degrees of airway obstruction, mucus accumulation, and airway wall remodeling [3]. The triggers for these pathologic changes are diverse, with important risk factors including parentage [4], type of feed/mode of feeding [5], and heat and pollen counts at the time of diagnosis [6]. Moreover, there are connections between EA and viral (i.e., equine influenza virus, equine herpesviruses, equine rhinitis viruses) or bacterial (i.e., *Streptococcus equi zooepidemicus*, *Actinobacillus* spp., *Pasteurella* spp.) diseases, which remain poorly understood [7,8]. While it is not clear whether these microbial etiologic agents incite the development of EA or are secondary colonizers of compromised airways, a history of infectious respiratory disease should be considered as a possible risk factor for EA in addition to other established causes.

Currently, the diagnosis of EA is multi-faceted, and the minimum database involves thorough history-taking, clinical scoring/presentation, lung function testing, examination of airway secretions collected via bronchoalveolar lavage (BAL) or tracheal wash (TW), and endoscopy [2]. Radiographs/ultrasound can be considered as supplemental diagnostics but are not considered part of the minimum database at this time, as they are neither sensitive nor specific in the diagnosis of EA [2]. All these diagnostic tools may not be necessary in the case of a horse with sEA in respiratory crisis, where history and clinical examination alone may be enough to make a preliminary diagnosis. However, for horses with mEA (or sEA in remission), a complete minimum database is often required. A history of episodes of reversible respiratory embarrassment precipitated by exposure to specific triggers (namely, moldy hay in the northeast of the United States and pasture allergens and particulates in the south) can often lead to a tentative diagnosis of sEA in remission. In contrast, mEA cases are often more difficult to diagnose; nonetheless, historical factors, such as genetics, current and lifetime exposures to particulates and allergens including feeds/feeding practices, barn environment, vaccinations, travel, and recent illness, should be considered. The Horse Owner Assessed Respiratory Signs Index (HOARSI) is one of the best-described questionnaire analysis tools for the classification of EA horses based on history [2]. However, clinical signs and indices are insufficiently sensitive to distinguish horses with mEA from normal horses or horses with sEA in remission [2].

Multiple clinical scoring systems have proven useful in distinguishing healthy horses from horses with sEA in crisis; most recently, the adapted 23-point scoring system was shown to be best at discriminating mild from severe EA, while the improved clinically detectable equine asthma scoring system (IDEASS) was shown to be the best at describing moderate to severe cases [2]. Unfortunately, as with the HOARSI, these scoring systems cannot differentiate horses with mEA from healthy horses or sEA horses in remission [2]. Interestingly, Gerber et al. found that “the subjective evaluation of airway disease treatment efficacy in a client-owned population of animals is better done by the owner than by a clinician during a single examination” [3]. Owners in that study were better able to detect changes in their horse’s performance, breathing efforts, coughing, and nasal discharge (as measured by a visual analog scale—VAS) than veterinarians. However, to the authors’ knowledge, no further studies have been published that consider the reliability of owner perception of clinical signs in EA.

Lung function testing is instrumental in understanding how a horse’s lungs are affected by EA at a specific moment in time. It is also the only tool that assesses how effective treatment has been at improving a horse’s mechanical ability to get air into and out of the lungs. Multiple modes of assessing lung function have been described in the literature to date. Esophageal balloon-pneumotachography (EB/P) is considered the gold standard for the measurement of airflow and pleural pressure. However, this technique is invasive in that it involves placing an esophageal balloon catheter to the level of the mid-thorax and connecting it to a pressure transducer, which allows for the calculation of pulmonary resistance, elastance, dynamic compliance, and pleural pressure. Open plethysmography (Open Pleth) is another way to evaluate lung function and is less invasive and more portable than the esophageal balloon method [9]. As with the esophageal balloon method, airflow is measured at the nose with a pneumotach. Rather than using a pressure transducer, Open Pleth uses elastic bands placed around the horse’s chest and abdomen to estimate a theoretical airflow at the chest. The true airflow at the nose is compared to the theoretical airflow at the chest, and the difference between the two values is used as a surrogate for respiratory resistance. Unfortunately, however, this technique is no longer commercially available. The most common lung function testing performed at the Tufts Cummings Hospital for Large Animals (HLA) is forced oscillatory mechanics (FOM). This method involves pushing pulses of compressed room air into a horse’s lungs and measuring the air as it comes back toward the pneumotach and pressure transducers. If a horse is experiencing bronchoconstriction or has mucus-filled airways, the pulses of air that return to the sensor will be smaller and asynchronous compared to those sent into the lungs. This information is used to calculate respiratory resistance. As with Open Pleth, this technique is not available commercially.

Lung mechanics measured using EB/P, Open Pleth, and FOM can be diagnostic in severely affected horses. However, these systems (especially the esophageal balloon method) are not sufficiently sensitive for demonstrating abnormal function in mildly affected horses whose baseline lung function is rarely abnormal. In these cases, bronchoprovocation (e.g., using histamine) or bronchodilation is used to distinguish between normal horses and horses with mEA [2]. While all horses will react to histamine with an increase in airway smooth muscle contraction and resultant increase in airway resistance, horses with EA or non-specific airway hyper-reactivity (AHR) respond to very small doses (<6 mg/mL) of aerosolized histamine with an increase in airway resistance, whereas horses without AHR require doses above 8 mg/mL to increase airway resistance. The dose required to increase resistance by at least 75% is known as the provocative concentration or PC75. When Open Pleth is used, a 35% decrease in delta flow at doses of histamine less than 6 mg/mL is used to establish a diagnosis of airway hyper-reactivity [10]. When horses have a baseline airway resistance above the normal population (above 0.8 cm H_2_O/L/s), bronchoprovocation is not performed; rather, the horse’s ability to dilate the airways in response to a bronchodilator drug (such as the beta-2 adrenergic agonist albuterol) is measured. This is termed a bronchodilator challenge.

Examination of respiratory secretions is an important method of diagnosis in EA, regardless of severity, because abnormal BAL fluid cytology (cell types or percentages) has been associated with poor performance and exercise intolerance [1]. BAL cytology in horses previously described as having IAD (mEA) is characterized by mild to moderate increases in neutrophil, eosinophil, and/or mast cell percentages, whereas BAL cytology in horses previously described as having RAO (sEA) is characterized by moderate to severe neutrophilia (>25% cells) and decreased lymphocyte and alveolar macrophage counts [1]. Tracheal wash (TW) may be a more practical way of sampling respiratory secretions in the field and has the additional benefit of allowing for bacterial culture. It can also be performed in cases where a BAL would be considered inappropriate (i.e., in horses with more severe respiratory compromise). However, as the tracheal cell population often differs from that of the lower airway (for instance, mast cells are rare in the trachea), tracheal cytology is more limited in its ability to reveal the lower airway cell milieu than BAL [2].

Endoscopic visualization of mucus can also aid in the diagnosis of EA. A mucus grade of 2/5 in racehorses or 3/5 in sport/pleasure horses, where 0 indicates an absence of mucus and 5 indicates a continuous stream of mucus through the length of the trachea, has been shown to be sufficient to predict mEA [2], although this continues to be debated [11]. Perhaps more importantly, however, endoscopy is critical to the detection of upper airway abnormalities in horses with mEA. Because the upper airways are the most important contributors to airway resistance, such abnormalities can alter the outcome of lung function testing or can even be the primary cause of cough and poor performance; two signs often ascribed to EA. Consequently, endoscopy should be routinely performed as part of the minimum database to rule out upper airway causes of obstruction as a primary cause of clinical signs or as a confounder of lung function testing.

Current treatment of EA is primarily aimed at controlling airway inflammation. Medical management is often implemented—with therapeutic choices often based on clinical experience—in combination with management of the environment. The common medications selected include systemic corticosteroids (e.g., dexamethasone, prednisolone) and inhaled corticosteroids (e.g., fluticasone, beclomethasone, budesonide, and ciclesonide) to address the primary problem of lower airway inflammation and inappropriate remodeling, as well as systemic bronchodilators (e.g., clenbuterol) and inhaled bronchodilators (e.g., albuterol, ipratropium bromide) to address the acute problem of bronchoconstriction. Additionally, inhaled mast cell-stabilizing drugs, such as cromolyn sodium, are often considered in horses with high BAL fluid mast cell counts [1]. Environmental management is focused on reducing exposure to airborne dust, which can be accomplished by using “low dust” feed/bedding, avoiding hay, improving ventilation in the barn, and increasing outdoor turnout [1].

In summary, although there are many researchers and clinicians who are actively investigating airway inflammation and EA, there is an apparent paucity of information regarding EA as a clinical disease. The 2016 ACVIM Revised Consensus Statement on inflammatory airway disease in horses recognized that “asthmatic horses of all severities have common clinical presentations (such as chronic cough, excess mucus, poor performance), but also a wide heterogeneity in terms of triggering factors, severity, and pathologic characteristics” [1]. This heterogeneity has made EA difficult to categorize. Clear episodes of reversible respiratory embarrassment seen at rest have come to define cases of “severe” EA (sEA), which has made sEA simpler to delineate and diagnose. Conversely, cases of “mild to moderate” EA (mEA) are more easily missed, since mEA, in its most subtle form, can cause exercise intolerance but no overt clinical signs of airway disease in horses performing in sports with high oxygen demands, such as racehorses [2]. Furthermore, there are no published data describing the longitudinal course of the disease, so there is no available information on baseline lung function stability over time. Therefore, the purpose of the study was to quantify and describe the types and range of clinical diseases that fall under the umbrella term of EA in horses in the New England area. In addition, this study considered the reliability of owners’ perceptions of their horses’ clinical signs and evaluated owner complaint scoring as a diagnostic tool. Lastly, this study sought to document and assess the diagnostic methods used at the Tufts Cummings School and the response of horses with EA to treatment and/or time.

## 2. Materials and Methods

### 2.1. Study Design

This study is a retrospective, cross-sectional, STROBE-compliant [12], observational analysis of equine patients who visited the Tufts Cummings Hospital for Large Animals (HLA) at least twice for evaluation and treatment of equine asthma (EA), also known as inflammatory airway disease (IAD) and/or recurrent airway obstruction (RAO) or heaves, from 1999 to April 2023. An initial search for patients who had at minimum either bronchoalveolar lavage (BAL) or pulmonary function testing (PFT) performed yielded 1012 independent records. The patients were further selected for inclusion of those who had at least two recorded visits to the HLA for evaluation of previously diagnosed or suspected EA, and whose medical records contained sufficient documentation of the presenting complaint, physical examination findings, bronchoalveolar lavage (BAL) and/or lung function data, and medication/treatment regimens (including relevant environmental management strategies). Patients with inadequate documentation, those who had only one visit recorded, or those who did not have equine asthma diagnosed as their primary respiratory disease were excluded (*n* = 936), leaving a final total of 76 horses included in this study (Figure 1).

### 2.2. Diagnostic Criteria

Mild/moderate equine asthma (mEA, previously referred to as inflammatory airway disease or IAD): Horses were classified as having mEA based on the criteria outlined in the most recent ACVIM consensus statement, which includes a history of poor performance or exercise intolerance, occasional coughing for more than a 3-weeks duration, bronchoalveolar lavage fluid (BALF) cytology characterized by mild increases in neutrophils (>5% and <25%), eosinophils (>1%), and/or mast cells (>2%), lack of visible respiratory effort at rest, and resting respiratory system resistance <0.6 cm H_2_O/L/s [2].

Severe equine asthma (sEA, previously referred to as recurrent airway obstruction/RAO or heaves): Horses were classified as having sEA if they showed the following combined clinical abnormalities: current or historic labored breathing at rest, chronic cough or nasal discharge, and evidence of disease chronicity based on anamnesis, with at least two episodes in the past and history of improvement when treated with corticosteroids, bronchodilators, environmental remediation, or a combination of these treatments [2].

### 2.3. History

Demographic information included age, breed, weight, feed, environmental management/housing, and presenting complaints (including cough, nasal discharge, nostril flare, respiratory effort, fever, and exercise intolerance). A patient’s presenting complaint was denoted as “recheck” only when their sole reason for visiting the HLA was a recheck visit and other prior complaints had been resolved. Visit dates were also logged so that the time to recheck could be assessed. Additionally, the history was examined to find evidence of the horse having been reported as “improved” qualitatively by the owner and/or quantitatively by the assessing veterinarian at Tufts.

### 2.4. Complaint Scoring

The complaint score focused on the client’s perception of the horse’s problem, rather than on the observations made by a veterinarian. The complaints were determined by examining the record using the written history and the communications log. Each complaint of cough, nasal discharge, nostril flare, respiratory effort (based on owner report), fever, and exercise intolerance was logged as a binary value of 1 (present) or 0 (absent) at each visit. Thus, only the presence of signs, and not severity, was assessed. Two versions of the complaint scores were generated using the sums of these complaints: a complaint score (CS) and an adjusted complaint score (ACS). The CS included the sum of all the complaints, whereas the ACS excluded exercise intolerance, since that variable was not directly comparable to the data collected upon physical examination. Thus, the maximum CS a horse could achieve was 6, and the maximum ACS a horse could achieve was 5. The minimum score for both CS and ACS was 0, which was associated with recheck visits where prior complaints had been resolved.

### 2.5. Medical Management

Details of recent historical and current treatments were cataloged at each visit date when available, focusing on the use of systemic or inhaled steroids and systemic or inhaled bronchodilators. Other pharmaceutical treatments as well as dietary supplements were noted whenever applicable. The pharmaceutical treatments prescribed at discharge were also recorded.

### 2.6. Clinical Examination

All the horses in the study underwent a thorough physical examination in which the following information was captured where available: respiratory rate, heart rate, rectal temperature, nasal discharge, nostril flare, cough, respiratory effort (based on abdominal effort and/or observation of heaves lines), and abnormal auscultatory findings. In cases for which the respiratory rate or heart rate was noted to be elevated but the rate was not documented, an elevated respiratory rate was defined as being >20 breaths/minute and an elevated heart rate was defined as being >44 beats/minute. Nasal discharge and coughing were recorded as being present when they occurred at rest or after an exercise challenge; the circumstances were not differentiated. Checking for inducible cough via gentle palpation of the larynx was a routine part of the physical examination process.

### 2.7. Clinical Examination Scoring

Each clinical examination finding of tachypnea, tachycardia, fever, cough, nasal discharge, nostril flare, respiratory effort, and abnormal lung sounds was logged as a binary value of 1 (present) or 0 (absent) at each visit. Because one goal of this work was to determine the relationship between owner complaints and findings upon examination by veterinarians at the HLA, two versions of examination scores were generated using the sums of these findings: an examination score (ES) and an adjusted examination score (AES). The ES included the sum of all the findings, whereas the AES excluded tachypnea, tachycardia, and abnormal lung sounds, since those variables were not directly comparable to the complaint data from the owners. Thus, the maximum ES a horse could achieve was 8, and the maximum AES a horse could achieve was 5. The minimum score for both ES and AES was 0.

### 2.8. Agreement Scoring

Agreement scores were calculated as a novel way to quantify the relationship between the complaint and examination scores. Two versions of the agreement scores were created for each visit. The first agreement score (AS) was calculated by subtracting the ES from the CS; therefore, the maximum AS achievable was 6 and the minimum was −8. The second adjusted agreement score (AAS) was calculated by subtracting the AES from the ACS; therefore, the maximum AAS achievable was 5 and the minimum was −5. The scores closest to 0 indicated a better agreement; positive scores indicated a worse agreement, where CS was greater than ES; and negative scores indicated a worse agreement, where ES was greater than CS.

### 2.9. Radiographs

Thoracic radiographs were performed in a subset of cases. Whenever radiographs were available, the findings were qualitatively scored as Grade 0 (normal), Grade 1 (mild changes), Grade 2 (moderate changes), Grade 3 (severe changes), and Grade 4 (abnormal structure present). This scoring was modified from Mazan et al. [13].

### 2.10. Endoscopic Examination

Endoscopic examinations were performed at rest and/or during exercise in a subset of cases. Any abnormalities of the respiratory mucosa, guttural pouches, or upper airway anatomy were described in detail, and a mucus score (0–4) was assigned based on the volume of respiratory secretions (RS) seen in the cranial thoracic trachea: Grade 0 (no RS visible), Grade 1 (small blebs of RS), Grade 2 (larger blebs of RS), Grade 3 (long strings of RS), and Grade 4 (significant strands/pools of RS). Grades 1 and below were accepted as normal [14].

### 2.11. Bronchoalveolar Lavage Fluid (BALF) Cytology

A bronchoalveolar lavage was performed using either a commercial cuffed BAL tube (Bivona) or with a 3 m bronchoscope, using 2 aliquots of 250 mL of warmed physiologic saline. The two samples were subsequently pooled, and slides were prepared using cytocentrifugation or sedimentation and pull-prep smears. The slides were stained with modified Wright stain and toluidine blue; the latter was used for the enumeration of mast cells. The cells (*n* = 500) were classified as the percentage of total cells that were macrophages, lymphocytes, neutrophils, eosinophils, and mast cells, and the presence of hemosiderin or other abnormalities (×1000 magnification). All the samples were read by a single individual (MRM) to ensure consistency across sample readings [15]. Designations of mild, moderate, and severe inflammation were made based on the following definitions: (1) mild inflammation: 5–19% neutrophils, 2- <3% mast cells, or 1- <2% eosinophils, (2) moderate inflammation: 20–40% neutrophils, 3–5% mast cells, or 2- <4% eosinophils, and (3) severe inflammation: >40% neutrophils, >5% mast cells, or ≥4% eosinophils.

### 2.12. Tracheal Aspirate

A small subset of horses with severe equine asthma (heaves) had respiratory embarrassment to the extent that it was deemed unsafe to perform a BAL. In these cases, an endoscopically guided tracheal aspirate (TA) was performed to acquire respiratory secretions for cytological analysis. These cytology data were excluded from our statistical analysis.

### 2.13. Lung Function Testing

Forced Oscillatory Mechanics: Mono-sinusoidal, multifrequency (1–3 Hz) forced oscillatory mechanics (FOM) was used to measure total respiratory system resistance (R_RS_), as previously described [16]. In short, FOM superimposes oscillations of compressed room air onto spontaneous breaths in awake horses to measure total R_RS_. The horses were sedated with 0.5 mg/kg xylazine intravenously (IV) and fitted with a latex-sealed low dead-space facemask, through which sinusoidal flow (generated using compressed air released through a proportional pneumatic valve) was superimposed over the horse’s spontaneous breathing. A pneumotachograph was used to measure the flow at the mask opening, and the difference between the mask and atmospheric pressures was recorded using a pressure transducer. For each frequency, coherence (signal-to-noise ratio) was calculated, with values >0.9 being accepted for analysis. After baseline measurements were obtained, changes in R_RS_ were used to monitor the effects of aerosolized histamine or bronchodilator challenge. During all measurements, the horse’s head rested on a stand or was physically supported by a holder in a neutral position. Note that abnormal baseline R_RS_ was defined as being ≥0.6 cm H_2_O/L/s.

Histamine Bronchoprovocation: Either histamine bronchoprovocation or albuterol bronchodilator challenge was performed to detect variability in airway caliber. Histamine bronchoprovocation was performed in horses with a baseline total R_RS_ below 0.80 cm H_2_O/L/s (within one standard deviation of the expected normal range), whereas albuterol bronchodilation challenge was performed in horses with an elevated R_RS_ as previously described [16]. Nebulization of 2 mL of 0.9% saline (negative control or saline baseline) over two minutes using a low dead-space face mask with a portable air compressor and nebulizer was performed and repeated for subsequent increasing concentrations of histamine diphosphate in saline solution (2, 4, 8, and 16 mg/mL). Lung function measurements were repeated following each nebulized dose until either a 75% increase in R_RS_ from the lowest previous measurement (FOM PC75 R_RS_) was achieved, or when the horse displayed a clinical reaction to histamine administration such as a notably increased respiratory rate or effort, or repeated coughing. Following FOM testing, a dose–response curve was generated to determine the histamine dose required to reach a 75% increase from the lowest baseline R_RS_ measured (PC75 R_RS_ for each horse). Airway hyperreactivity was classified as a PC75 R_RS_ less than or equal to 6.0 mg/mL histamine. For albuterol bronchodilation, 1–2 µg/kg of albuterol sulfate, rounded to the higher dose, was administered using the Aerohippus delivery device as previously described. Fifteen minutes were allowed for the horse to develop maximum bronchodilation, then R_RS_ was again measured using FOM.

Open Plethysmography: In a subset of cases, pulmonary function testing and histamine bronchoprovocation were performed using a commercial flowmetric plethysmography system, as described by Wichtel et al. [10]. When Open Pleth is used, a 35% decrease in delta flow at doses of histamine less than 6 mg/mL is used to establish a diagnosis of airway hyper-reactivity [10].

### 2.14. Statistics

The data were assessed for normality, first using histograms for visual assessment and then Kolmogorov–Smirnov tests for normality. For the data that were continuous and normally distributed, descriptive statistics were performed, with the central tendency and variability shown as the mean and standard deviation, respectively. For ordinal data, or continuous data that were non-normally distributed, the central tendency and variability are shown as the median and range. McNemar tests were used to compare paired, nominal dichotomous data from visits 1 and 2. Wilcoxin ranked sums were calculated for paired ordinal data and used to compare variables between 2 groups. Paired *t*-tests were used to evaluate interval data. Chi-square tests of independence were performed to examine (1) the relationship between various owner complaints and physical examination findings and (2) the relationship between mEA and sEA physical examination findings. All the data were evaluated using commercially available statistical software (SPSS Statistics Version 28.0.1.1 [14] and Excel Version 16.72 (23040900)).

## 3. Results

### 3.1. Demographics and Visit Description

In total, 76 horses were included in this study. The most common types of horses were Quarterhorses (31.5%), Warmbloods (17.1%), and Thoroughbreds (9.2%). Of the remaining horses, 14.5% were mixed breed, 3.9% were of unknown breeds, and 23.7% were from breeds with four or fewer individuals represented (including Morgans, Draft horses, Ponies, Icelandics, Saddlebreds, Standardbreds, and Arabians). When separated by diagnosis, the breeds most commonly noted with both mEA and sEA horses were Quarterhorses (33.3% and 22.2%, respectively). Warmbloods (18.5%) and mixed breeds (16.7%) were also commonly represented within the mEA group. For sEA horses, Warmbloods, Morgans, and horses of mixed and unknown breeds each represented by 11.1%. Notably, four individuals had varying diagnoses across visits—for example, a horse may have been given a diagnosis of mEA at visit 1 but may have been re-classified as sEA by visit 2. The median age of all the included horses was 14.0 years old (range of 2–34 years old). When separated by diagnosis, the median age of the mEA horses was 14.0 years old (range of 2–30 years old), whereas the median age of the sEA horses was 17.0 years old (range of 7–34 years old). Of all the included horses, 48/76 (63.2%) were geldings, whereas 28/76 (36.8%) were mares. Of the horses given an eventual diagnosis of exclusively mEA, 37/54 (68.5%) were geldings and 17/54 (31.5%) were mares. Comparatively, of the horses given an eventual diagnosis of exclusively sEA, 7/18 (38.9%) were geldings and 11/18 (61.1%) were mares. Further demographic information and characteristics of included cases can be found in Table 1 below.

Only 46/76 (60.5%) of the horses had their weights recorded. These horses were separated into weight categories, where “small” was defined as <400 kg, “average” was defined as 400–600 kg, and “large” was defined as >600 kg. Thirty-two horses were categorized as average, ten were categorized as large, and four were categorized as small. The mean weight was found to be 502.2 kg (SD 101.9).

There were a total of 197 visits included in this study, 71 of which were considered “recheck” visits, meaning any prior owner complaints had been resolved. The years with the most included initial and repeat visits were 2002 and 2019 (11 and 10 initial visits and 14 and 19 repeat visits, respectively). The mean number of initial visits per year was 3.2 (SD 2.9), whereas the mean number of repeat visits per year was 5.0 (SD 4.2). A total of 68.4% of the horses had one repeat visit (two visits total), 18.4% of the horses had two repeat visits, and 7.9% of horses had three repeat visits. The most repeat visits any individual horse had was seven (*n* = 1). The mean number of days between subsequent visits was found to be 302 (SD 487). Of these visits, 68 were found to occur greater than 2 months after the previous visit, 31 were found to occur between 2 weeks and 2 months after the previous visit, and 22 were found to occur within 2 weeks of the previous visit. Of the 197 visits, 138 (70.0%) resulted in a diagnosis of mEA and 45 (22.8%) resulted in a diagnosis of sEA [2]. Nine of the 138 mEA diagnoses (6.7%) showed a concurrent finding of exercise-induced pulmonary hemorrhage (EIPH).

### 3.2. Agreement between Owner Complaint Versus Veterinary Physical Examination Findings

The owner complaints included nasal discharge, nasal flare, respiratory effort, fever, and exercise intolerance. Across all visits, the most common owner complaint was cough (74 instances), followed by respiratory effort (62 instances), exercise intolerance (62 instances), and nasal discharge (26 instances). There were only seven instances where an owner reported nasal flare and only one instance where an owner reported fever. Each complaint of cough, nasal discharge, nostril flare, respiratory effort, fever, and exercise intolerance was logged as a binary value of 1 (present) or 0 (absent) at each visit. Excluding recheck visits where no complaints were logged, the mean complaint score (CS) (comprising the sum of the complaints) was found to be 1.8 (SD 0.8). Therefore, the average number of complaints per visit was ~2.

The physical examination findings that were recorded during visits to the HLA included respiratory rate, heart rate, rectal temperature, nasal discharge, nostril flare, cough, respiratory effort, and abnormal auscultatory finding. Across all visits, the most common examination finding was respiratory effort (47 instances), followed by nasal flare (46 instances), abnormal auscultatory finding (37 instances), nasal discharge (33 instances), and cough (27 instances). Only one horse was febrile upon physical examination; this instance was in agreement with the owner’s complaint of fever above. Sixty-seven instances of tachypnea were logged compared to 111 instances of eupnea; however, the mean respiratory rate was found to be tachypneic, at 24 breaths per minute (SD 10.2). Only 10 instances of tachycardia were recorded compared to 166 instances of normal heart rate. The mean heart rate was 38 beats per minute (SD 8.0). Each clinical examination finding of tachypnea, tachycardia, fever, cough, nasal discharge, nostril flare, respiratory effort, and abnormal lung sounds was logged as a binary value of 1 (present) or 0 (absent) at each visit. The mean examination score (ES) (comprising the sum of the findings) was 1.4 (SD 1.5). Therefore, the average number of abnormal physical examination findings per visit was between 1 and 2.

Agreement scores were calculated to quantify the relationship between the complaint and examination scores. Two versions of agreement scores were created for each visit. The first agreement score (AS) was calculated by subtracting the ES from the CS. The second adjusted agreement score (AAS) accounted for only those complaint and examination variables which could be directly compared (cough, nasal discharge, nasal flare, respiratory effort, and fever) and was calculated by subtracting the adjusted ES (AES) from the adjusted CS (ACS). The mean AS was calculated to be −0.18 (SD 1.5), while the mean AAS was calculated to be 0.08 (SD 1.2).

Additionally, Chi-square tests of independence were performed to specifically compare the individual components of the AAS (excluding fever) for the mEA and sEA cases separately (Table 2). Regardless of EA disease severity, it was significantly more likely for the owners to report cough than it was for cough to be noted during the veterinary examination, and it was significantly more likely for nasal flare to be noted upon veterinary examination than it was for it to be reported by the owners. Interestingly, for the mEA horses, it was significantly more likely for the owners to report respiratory effort than it was for respiratory effort to be noted during the veterinary examination, whereas the sEA horses demonstrated the opposite pattern. It is important to consider that: (1) this study was largely unable to tell whether the clients were referring to their horses’ respiratory effort at rest or during exercise, and (2) respiratory effort was appreciated on the veterinary physical examination via the presence of abdominal effort or heave lines. For all the horses included in this study, regardless of EA severity, there was no significant difference regarding the likelihood of nasal discharge being reported by the owner versus being detected upon veterinary examination.

The physical examination findings were also compared in the horses given an eventual diagnosis of mEA and sEA via Chi-square tests of independence (Table 3). Nasal discharge, nasal flare, respiratory effort, abnormal lung sounds, and tachypnea were all significantly more likely to be noted on the veterinary examinations in the horses with sEA compared to the horses with mEA. Cough was the only physical examination finding that did not have a significant difference in the mEA versus sEA horses.

### 3.3. Other Components of the Clinical Examination

Radiographs were performed on 60/197 visits (30.5%), representing 43/76 of the horses (56.6%). Changes described as “mild” were the most common (45.8%), followed by radiographs with no abnormal findings documented (27.1%). Radiographic changes described as “moderate” and “severe” were much less common (13.6% and 5.1%, respectively). 8.5% of the radiographs were said to include “abnormal structures.”

Endoscopic examination was performed on 82/197 visits (41.6%), representing 48/76 of the horses (63.2%). Mucus scores (0–4) were assigned based on the volume of respiratory secretions (RS) seen in the cranial thoracic trachea, with grades 0 and 1 accepted as normal. The most frequent mucus score assigned was grade 2 (37.8%), followed by grade 1 (31.7%). All other mucus scores were assigned in <13% of cases each.

A bronchoalveolar lavage (BAL) was performed in 130/197 of the visits (66.0%), representing 64/76 of the horses (84.2%). Thirty-five (54.7%) of these horses had repeat BALs performed at subsequent visits. The mean (SD) BAL percentage of neutrophils (PMN) was 21.7 (18.1) at the first visit and 16.0 (17.5) at the second visit. This difference was not significant, with a confidence interval of [−10.1, 11.4], and *p* = 0.058. At the third visit, the mean (SD) BAL percentage of neutrophils was 25.78 (21.3); this difference was again not significant, with a CI of [−23.5, 7.8] and *p* =0.79. The mean (SD) BAL percentage of mast cells was 2.45 (1.95) at the first visit and 2.84 (2.74) at the second visit. Similarly, this difference was not significant, with a CI of [−1.61, 0.81] and *p* = 0.880. At the third visit, the mean (SD) BAL percentage of mast cells was 2.86 (4.14). The difference from visit 2 was not significant, with a CI of [−2.92, 2.60] and *p* = 0.774. Comparisons were not made beyond the third visit because most of the horses included in the study had a maximum of three visits. Of all 130 instances where BAL was performed, 31 (23.8%) were considered to show severe inflammation, 51 (39.2%) were considered to show moderate inflammation, and 45 (34.6%) were considered to show mild inflammation. Three (2.3%) of the included BALs did not meet any criteria for airway inflammation. A total of 54/130 of the BAL samples (41.5%) showed neutrophilic inflammation alone, 26/130 of the BAL samples (20.0%) showed mast cell inflammation alone, and 2/130 of the BAL samples (1.5%) showed eosinophilic inflammation alone. Excluding the three samples which did not meet any criteria for airway inflammation, the remaining 45/130 samples (34.6%) showed mixed inflammation.

Lung function testing using FOM was performed on 167/197 of the visits (84.8%), representing 70/76 of the horses (92.1%). Sixty-two (88.6%) of these horses had repeat lung function testing performed on subsequent visits. Of all the 167 instances where lung function was assessed using FOM, there were 64 instances (38.3%) where the horse being examined was found to have an abnormal baseline total respiratory system resistance (R_RS_), compared to 103 instances (61.7%) where the horse being examined was found to have a normal baseline R_RS_. Histamine bronchoprovocation was performed in 142 cases, 93 (65.5%) of which were found to be definitively abnormally reactive. Comparatively, 29.6% of the cases were found to be normally reactive (PC75 R_RS_ greater than or equal to 8.0 mg/mL histamine). One case was considered normally reactive based on a PC100 R_RS_ greater than or equal to 8.0 mg/mL histamine, because no PC75 R_RS_ was recorded in that instance. Seven cases (4.9%) were classified as ambiguous based on their PC75 R_RS_ being greater than 6.0 but less than 8.0 mg/mL histamine.

An additional four visits (representing four separate horses) used Open Pleth as a method of non-invasive lung function testing. Three of these found the horses to be abnormally reactive based on a PC35 and/or PC50 R_RS_ less than or equal to 6.0 mg/mL histamine.

There were 16 instances where histamine bronchoprovocation was not performed and albuterol bronchodilation (or ipratropium bronchodilation in 3 cases) was pursued instead. The mean bronchodilator percent improvement in R_RS_ was found to be 28.8% (SD 17.9).

### 3.4. Pharmaceutical Treatments at Intake

Fewer than 50% of the horses were being treated with any medication for suspected asthma at intake for the first visit. Of those horses already being treated, the most commonly used medications for mEA and sEA horses were systemic steroids (33.3% and 36.1%, respectively), followed by inhaled bronchodilators (16.7% and 19.4%, respectively). The horses with sEA had been prescribed inhaled steroids and systemic bronchodilators equally (16.7% each), whereas the horses with mEA had been prescribed more systemic bronchodilators (13.9%) compared to inhaled steroids (5.6%). The mEA horses had also been prescribed a greater number of “other” medications at intake (30.6%) compared to the sEA horses (11.1%). These medications included antibiotics, antihistamines, mast cell stabilizers, and vitamin supplements.

### 3.5. Pharmaceutical Treatments at Discharge

At discharge after the first visit, both mEA and sEA horses were prescribed inhaled steroids more than any other drug class (34.6% and 44.0%, respectively). The next most-common medication the mEA horses were discharged with was systemic steroids (29.6%), whereas the sEA horses were discharged with more inhaled bronchodilators (32.0%) than systemic steroids (20.0%). The mEA horses were discharged with inhaled bronchodilators only 24.7% of the time. Neither the mEA nor sEA horses were discharged with any systemic bronchodilators. As at intake, the mEA horses were discharged with a greater number of “other” medications than the sEA horses (11.1% versus 4.0%). Of the “other” medications prescribed to the mEA horses, 6/9 (66.7%) were mast cell stabilizers. The remaining three medications included one each of thyroxine, phenylbutazone (prescribed due to a concurrent dental floatation), and intranasal phenylephrine. The only “other” medication prescribed to a horse with sEA was trimethoprim-sulfa antibiotic (*n* = 1). On at least one visit each, 7/76 of the horses (9.2%)—all diagnosed with mEA—were not discharged with any medications. Four of those seven horses were explicitly (within our records) given recommendations for environmental remediations, including switching to a low-dust feed.

### 3.6. Assessing Improvement over Time

In cases where improvement was explicitly assessed by the veterinarian compared to the last visit, there were 66 instances of improvement and 17 instances where the horses were deemed to not have improved. Similarly, of the instances when the treatment response was explicitly assessed by the owner compared to the last visit, there were 37 instances of improvement and 16 instances where the horses were deemed to not have improved. Using a McNemar test for paired dichotomous variables, an owner’s complaint of cough was found to be significantly less likely at the second visit compared to the first. Considering only the cases where the veterinary and owner input was collected, there were 30 instances of veterinary–owner agreement and 13 instances of disagreement (6 of which where the veterinarian deemed a horse improved while the owner disagreed and 7 of which where the owner deemed a horse improved while the veterinarian disagreed).

To further assess improvement over time, adjusted complaint scores (ACS) and adjusted examination scores (AES) were compared at visit 1 versus visit 2. Using the Wilcoxon Signed Ranks Test, there was a significant decrease in both the computed client ACS and the computed veterinary AES. Additionally, the PC75 R_RS_ was compared between visits using a paired *t*-test and was found to be significantly increased at visit 2 compared to the initial visit 1—increasing from 4.6 to 7.7 mg/mL histamine—with a higher PC75 R_RS_ indicating less reactive airways. No significant differences were found in BAL fluid cell proportions over time.

## 4. Discussion

This study presented a retrospective analysis of a subset of equine patients who visited the Tufts Cummings HLA at least twice for evaluation and treatment of EA from 1999 to April 2023. The goals were: (1) to quantify and describe the types and range of clinical diseases that fall under the umbrella term of EA in horses in this geographic area, (2) to consider owners’ perceptions of their horse’s clinical signs and evaluate owner complaint scoring as a diagnostic tool, and (3) to document and assess the diagnostic methods used at the Tufts Cummings School and assess the response of horses with EA to treatment and/or time.

One key finding from this study is that owners can be reliable evaluators of their asthmatic horses. Although the mean calculated AS was −0.18, which points to veterinarians being at least mildly more perceptive than owners, when the owner complaints were compared against only the same veterinary physical examination findings (cough, nasal discharge, nasal flare, respiratory effort, and fever) via the AAS, the mean calculated score was 0.08. This adjusted score is a much fairer—and likely a more accurate—comparison of owner complaints relative to veterinary findings. In the original AS, the veterinarian has a greater possible number of points that could be allocated, but the AAS accounts for and balances this bias. The AAS of 0.08 means that, on average, owners and veterinarians are in agreement on a patient’s clinical picture. The fact that the AAS is even slightly positive raises the question of whether owners, who have the benefit of more observations, were actually more reliable observers than the veterinarians. When individual components of the AAS were evaluated using Chi-square tests of independence, it was determined that the answer to this question is probably yes. The owners were significantly more likely to report coughing in their asthmatic horses, regardless of disease severity, than the veterinarians were to note coughing upon physical examination. This finding aligns with findings by Gerber et al., who concluded that the owner who sees the horse frequently is better able to assess the effects of treatments on cough than the clinician who sees the horse only during a brief examination [3].

Interestingly, the owners of mEA horses were also more likely than the veterinarians to discern respiratory effort, although the owners of sEA horses were not. As horses with mild asthma, by definition, have no respiratory effort at rest, it may be that the owners actually referred to the increased respiratory effort during or shortly after exercise that these horses often experience. Respiratory effort in severely affected horses is commonly assessed by veterinarians by the degree of abdominal effort or the presence of hypertrophied expiratory muscles (heave lines). More clinically affected sEA horses are more likely to present with these outwardly visible symptoms of disease than subclinical mEA horses. Consequently, while veterinarians may be able to accurately assess respiratory effort in sEA horses, if an mEA horse is examined only at rest, no signs of respiratory effort will be seen. This study was largely unable to tell whether the clients were referring to their horses’ respiratory efforts at rest or during exercise. Furthermore, the sEA horse owners may be worse at detecting respiratory effort in their horses compared to the mEA horse owners, because their horses are so clinically affected that they are no longer regularly exercising them. It is also interesting to speculate that the owners may be so accustomed to their horses’ breathing that they no longer perceive it as abnormal. Lastly, horses with sEA have a different breathing strategy than normal horses [17]. Rather than demonstrating the normal biphasic inspiration and expiration typical of healthy horses at rest, horses with sEA have a constant inspiratory and expiratory flow throughout the duration of their tidal volume [17]. This may also make it difficult for owners of sEA horses to judge their respiratory effort, since sEA horses have more consistently inadequate breathing with less variability in each breath.

Nasal flare, unlike cough and respiratory effort, was reported significantly more often by the veterinarians than it was by the owners. Veterinarians are trained and routinely examine normal and affected animals; thus, it is not surprising that they are better at discerning a normal amount of nostril movement during breathing as compared to flare. On the other hand, owners of asthmatic horses (who are not necessarily seeing healthy animals regularly) may grow accustomed to their horses’ nostril movements. These owners may lose the ability to discriminate between what is normal and what should be considered nasal flare, and so would be less likely to report nasal flare as a complaint.

The lack of a statistically significant difference between nasal discharge reported by the owners versus by the veterinarians upon physical examination can be explained because so few mEA and sEA horses were reported to have any discharge at all. This is interesting, given the fact that both coughing and nasal discharge are often offered as common—although nonspecific—signs of EA [2]. In this study population, however, nasal discharge was far less common in horses with EA than the literature suggests. Regardless, when the sEA horses were compared to the mEA horses upon physical examination, the sEA horses were found to have significantly more nasal discharge (in addition to more nasal flare, respiratory effort, abnormal lung sounds, and tachypnea). This is unsurprising, given that horses with severe asthma are more clinically affected than their mild–moderate counterparts. What is interesting, however, is the finding that there is no significant difference in coughing seen upon physical examination of sEA versus mEA patients. Coughing tends to be intermittent in mEA; thus, veterinarians may be less likely to witness it upon examination of these patients [3]. Yet, even in sEA, where coughing is commonly more frequent (if not regular), veterinarians are not more perceptive.

Notably, of the 76 horses included in the study and across 197 visits, only 1 horse ever presented with a fever. This aligns with the current understanding of EA and the consensus that most affected horses do not have a fever unless a secondary respiratory infection has occurred [18].

Another important finding from this study is that while airway reactivity (PC75 R_RS_) and clinical signs improved in a majority of these horses—with AES, ACS, and PC75 R_RS_ decreasing at visit 2 relative to visit 1, due either to treatment or to time—BAL fluid cell inflammatory cell proportions did not. There is disagreement in the current literature regarding the relationship between BAL cytology and lung function testing/histamine bronchoprovocation: while some studies have found moderate to strong correlations between the two [19,20,21], others have not [10,22]. It is apparent that some individuals may have abnormal lung function and normal BAL cytology or vice versa, which is why both are part of the minimum database used to diagnose EA [2]. While treatment with corticosteroids often effects a noticeable change in the horse’s breathing effort and can result in improved lung function, only environmental remediation appears to have a robust association with improvement in BAL cytological evidence of airway inflammation. According to the 2016 ACVIM consensus statement on IAD, “there is only nonpeer-reviewed evidence that dexamethasone and fluticasone are effective in decreasing airway hypersensitivity and reactivity in IAD horses and in both cases the BAL cytology was not significantly affected” [1]. Furthermore, “the lack of decrease in BAL neutrophil percentages after short-term glucocorticoid treatment has … been observed in several RAO studies where the air quality was kept unchanged, [and] one study found that long-term dexamethasone and fluticasone treatment did not reverse the airway neutrophilia when RAO horses are kept indoors and exposed to hay, even after 6–7 months” [1]. To the authors’ knowledge, no studies exist that truly compare lung function testing/histamine bronchoprovocation and BAL cytology as a means to monitor EA improvement longitudinally. Consequently, future research should focus on verifying whether lung function testing is reliably a better indicator of EA improvement than BAL fluid cytology. Finally, in the majority of cases, both the veterinarian and the owner agreed about the degree of improvement seen. The few instances of owner–veterinarian disagreement may be explained by the veterinarian’s ability to incorporate all diagnostic outcomes, such as lung function testing and BAL cytology, whereas the owners must rely upon the horse’s clinical appearance and performance.

The information about pharmaceutical treatment of EA that emerged from this study is also relevant. The fact that both mEA and sEA horses are prescribed inhaled steroids more than any other drug is not surprising, given that both inhaled and systemic corticosteroids have been shown to improve lung function in sEA horses [23]. Although a naturally occurring model of mEA in Thoroughbred polo ponies exposed to bushfire smoke showed no superiority of dexamethasone treatment over a placebo, with both groups improving with environmental modification [24], the empirical treatment of mEA horses using corticosteroids is common and often clinically successful. It is also important to consider, however, that inhaled steroids being the most commonly prescribed drug in this study may represent a treatment bias, as these data only reflect the treatment regimen employed by one hospital. The finding that sEA horses are much less frequently prescribed systemic steroids than mEA horses is interesting, however. It is possible that this difference represents the reported demographic differences of typical mEA versus sEA horses, as sEA horses tend to be older, whereas mEA horses are usually younger or middle-aged [1]. This demographic difference is reflected in our study population, where the mean age of the mEA horses was 13.9 years old (range of 2–30 years old) and the mean age of the sEA horses was 17.1 years old (range of 7–34 years old). The decreased prescription of systemic steroids in the sEA horses may therefore reflect the clinicians’ wariness to prescribe these medications to patients who may be more predisposed to (or more likely to already have) concurrent diagnoses that make them more sensitive to the side effects of systemic steroids.

It is also interesting to note that no systemic bronchodilators were prescribed to the EA horses after their first visit at the Tufts Cummings School, regardless of disease severity, whereas inhaled bronchodilators were commonly prescribed to both groups of patients (but especially the sEA horses). It is likely that the inhaled bronchodilators (e.g., albuterol) were often prescribed as “rescue” medications due to their rapid onset of action (5 min) and ease of administration [25]. This may negate the necessity for systemic bronchodilators, at least at the first visit. Furthermore, clenbuterol has been shown to cause tachyphylaxis with prolonged use. Read et al. demonstrated that although clenbuterol initially lowered airway sensitivity to inhaled histamine, tachyphylaxis that resulted in increased airway reactivity was evident by day 21 [26]. Similar tachyphylaxis did not occur over a 2-week period of use in horses with mEA [27]. In human asthmatics, the concomitant use of a corticosteroid along with a bronchodilator reduces this tachyphylaxis [28], which may explain why steroids are often selected as a first-line medication. Additionally, it is important to remember that bronchodilators are not benign drugs—an overdose may result in sinus tachycardia, muscle tremors, hyperhidrosis, and colic [29]. Since albuterol must be inhaled and has local effects which subside relatively quickly, the likelihood of a severe overdose is much less than it is for clenbuterol. For all these reasons, veterinarians may be more inclined to prescribe a local rather than a systemic bronchodilator.

Lastly, the fact that the mEA horses were prescribed more “other” medications than the sEA horses—and that these “other” medications were predominantly mast cell stabilizers—was not unexpected. It is logical that mast cell stabilizers would be given to mEA horses, who may have excessive mast cells in their BAL fluid, as opposed to sEA horses, who tend to have a more purely neutrophilic inflammation.

This study has unavoidable, yet important limitations. Firstly, the information used in the study was collected over a 24-year period. The quality of assessment of EA has changed greatly in that time, both due to improving technology and enhanced record-keeping. There were some horses included (particularly older horses who visited HLA in the early years of the study) who had considerable gaps in their records either because the information was not collected, or because it was lost over time. There was also no way to ensure the information was collected in the same manner across individual cases; physical examinations were performed by different clinicians, and different questionnaires were used for history-taking amongst the owners. It is possible, for example, that certain owner complaints or physical examination findings were missed because the interviewer failed to ask, the client failed to mention it, and/or there were sparse physical examination records. It is also important to consider that this study selected for patients with >1 visit, which may have introduced certain biases into the data. Specifically, the degree of improvement with treatment and/or time could be underestimated here, as the patients with >1 visit may have returned because they were less likely to have clinically improved. However, 18/76 horses (23.7%) had a second visit scheduled to participate in EA-related clinical trials (not connected with this study), and 15 of those 18 horses were returning with no remaining owner complaints (i.e., returning for “recheck” visits). Unsurprisingly, the years with the most included initial and repeat visits, 2002 and 2019, were both years in which there was active recruitment for participation in EA-related clinical trials. Furthermore, in the first 4 years of the study period, repeat visits were performed for free to incentivize clients to return—this may have also influenced the pattern of visits over time. Lastly, since this study involved predominantly descriptive statistics, it has limited scope in that it cannot establish any cause-and-effect relationships. Further research is needed to substantiate many of the findings presented here. Nonetheless, the information included in this study is extensive and contains a greater volume of longitudinal data on EA than has ever been published to the authors’ knowledge.

## 5. Conclusions

Our data reveal that owners can be astute evaluators of their horses’ clinical conditions over time, and that there are symptoms for which owners are in fact more reliable than veterinarians (such as detecting cough in mEA and sEA patients and respiratory effort in mEA patients). This is likely because owners often have more observational data on their horses. The implications of this are important if it can be understood and leveraged in “team-based” veterinary healthcare. This study also demonstrates that a majority of EA patients who came to and were treated at the Tufts Cummings School improved with treatment and/or time, and that this improvement may be monitored using veterinary AES and PC75 R_RS_ in addition to owner-determined ACS. Specifically, owner complaints of cough may be an especially reliable way to track EA improvement and may in fact be more useful at tracking EA improvement longitudinally than BAL cytology. Furthermore, this study provided a substantial description of the types and range of clinical diseases that fall under the umbrella term of EA in horses in this geographic region. It also illustrated the largely positive effects of treatment, time, or both, on clinical signs, examination of the respiratory system, and lung function testing.

## Figures and Tables

**Figure 1 animals-13-03387-f001:**
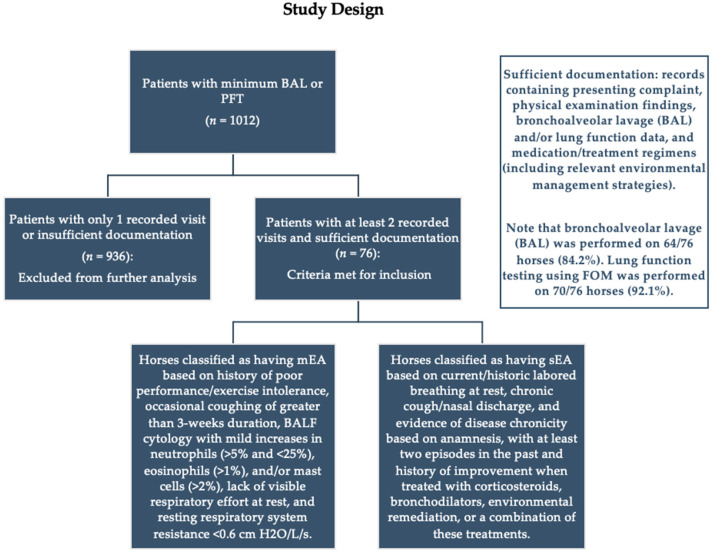
Study Design. This figure describes patient selection for inclusion in the study. Classifications of mEA and sEA were based on the criteria outlined in the most recent ACVIM consensus statement [2].

**Table 1 animals-13-03387-t001:** Characteristics of included cases, stratified by mEA and sEA diagnoses. Note that not all visits resulted in definitive mEA or sEA diagnoses (some cases were diagnosed at previous/subsequent visits). Therefore, the “mEA cases” and “sEA cases” columns will not consistently sum to the “All cases” column.

Sample Characteristic	All Cases (*n* = 76)	mEA Cases (*n* = 54)	sEA Cases (*n* = 18)
Demographics			
^1,2^ Sex (frequency)			
Mare	28 (36.8%)	17 (31.5%)	11 (61.1%)
Gelding	48 (63.2%)	37 (68.5%)	7 (38.9%)
Age, years (median [range])	14 (32)	12 (28)	17 (27)
Age, years (frequency)			
2–10	42	36	2
11–19	37	74	27
20–28	148	24	13
>28	4	1	3
^1,2^ Breed (frequency)			
Arabian	1 (1.3%)	0 (0%)	1 (5.6%)
Crossbred	11 (14.5%)	9 (16.7%)	2 (11.1%)
Draft horse	3 (3.9%)	2 (3.7%)	1 (5.6%)
Icelandic	3 (3.9%)	2 (3.7%)	1 (5.6%)
Morgan	4 (5.3%)	2 (3.7%)	2 (11.1%)
Pony	3 (3.9%)	2 (3.7%)	1 (5.6%)
Quarterhorse	24 (31.6%)	18 (33.3%)	4 (22.2%)
Saddlebred	2 (2.6%)	1 (1.9%)	1 (5.6%)
Standardbred	2 (2.6%)	1 (1.9%)	1 (5.6%)
Thoroughbred	7 (9.2%)	6 (11.1%)	1 (5.6%)
Unknown/missing	3 (3.9%)	1 (1.9%)	2 (11.1%)
Warmblood	13 (17.1%)	10 (18.5%)	2 (11.1%)
Visits (frequency)			
All visits	197	135	45
Initial visits	76	48	21
Repeat visits	121	87	24
Recheck visits (prior owner complaints resolved)	71	49	16
Owner complaints (frequency)			
Cough	74	48	21
Febrile	1	1	0
Nasal discharge	26	17	8
Nostril flare	7	2	5
Respiratory effort	62	36	20
Veterinary examination findings (frequency)			
Cough	27	18	9
Febrile	1	1	0
Nasal discharge	33	18	14
Nostril flare	46	23	20
Respiratory effort	47	16	30
BAL testing (frequency)	130	94	28
Lung function testing (frequency)	167	121	34
Medications prescribed after visit 1 (frequency)			
Inhaled bronchodilators	28	20	8
Systemic bronchodilators	0	0	0
Inhaled steroids	39	28	11
Systemic steroids	30	24	5
Other medications	11	9	1

^1^ Horses who received conflicting mEA and sEA diagnoses were excluded from these sections’ mEA and sEA columns. Therefore, the “mEA cases” and “sEA cases” columns will not consistently sum to the corresponding “All cases” column. ^2^ In these sections, the frequencies are describing the number of individual horses and not the number of visits. Therefore, a percentage is given relative to the number of individuals in each column (*n*).

**Table 2 animals-13-03387-t002:** Chi-square analysis of owner complaint versus veterinary physical examination findings in mild to moderate EA and severe EA. *p* values with asterisks represent statistically significant findings. Values highlighted in grey indicate variables more frequently observed by owners than by veterinarians.

		*X*^2^ Statistic	*p* Value
Mild to Moderate EA	Cough	18.0481	0.000022 *
	Respiratory effort	19.44	0.00001 *
	Nasal flare	19.44	0.00001 *
	Nasal discharge	0.0382	0.856225
Severe EA	Cough	7.2	0.00729 *
	Respiratory effort	4.5	0.003895 *
	Nasal flare	12.4615	0.000415 *
	Nasal discharge	2.1658	0.14113

**Table 3 animals-13-03387-t003:** Chi-square analysis of veterinary examination findings in mEA versus sEA horses. *p* values with asterisks represent statistically significant findings.

Physical Examination Findings	*X*^2^ Statistic	*p* Value
Cough	1.1765	0.278076
Respiratory effort	19.6485	<0.00001 *
Abnormal lung sounds	29.4954	<0.00001 *
Tachypnea	29.4954	<0.00001 *
Nasal discharge	7.2973	0.006906 *
Nasal flare	13.9433	0.000188 *

## Data Availability

Data are available from the corresponding author upon request.

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
