# Peer review of "A Longitudinal Analysis of Equine Asthma Presentation and Response to Treatment Using Lung Function Testing and BAL Cytology Analysis in Combination with Owner Perception"

_animals, 2023, doi:10.3390/ani13213387_

Round 1

Reviewer 1 Report

Comments and Suggestions for Authors

This paper reports various findings arising from the investigation of horses making multiple visits to a specialist clinic for evaluation of equine asthma. As the authors state, equine asthma is an important clinical problem particularly when signs in afflicted horses are subtle. For this reason, the decision to undertake the study has scientific validity.

The approach used to collect the data presented is appropriate given the outcomes the authors are trying to achieve.

Introduction

The introduction is well written and provides an excellent overview of the topic of equine asthma. It highlights the paucity of studies assessing the course of the disease longitudinally. If necessary, the content of the introduction could be reduced but I think the content as presented is very useful, particularly for people who may not be particularly familiar with the topic under discussion.

Materials and methods

The materials and methods give an appropriate description of the data that were collected. The information provided would enable the study to be replicated at other centres if the required information were available.

The statistical methods used to analyse the data are appropriate.

Results

The results are clearly presented and complete.

It would be appropriate to include standard deviations with the means.

It would be interesting to know the uses of the horses , although I acknowledge that this data may not be available. For example, 44% of horses were from breeds which are commonly used for high intensity exercise (quarterhorse, thoroughbred standardbred). Where these horses undergoing race training, in which case they would probably be stabled, or were they retired? What percentage of the other horses were used for athletic pursuits? As stated above this is not essential, and the data may not be available, but it could be of value to discuss these details if possible.

Discussion

The discussion is appropriate and highlights the important findings of the study.

Overall Comment

There are inherent limitations in studies of this type but they can provide useful information for practitioners. I think this is a very useful study and that the paper is very well written. I commend the authors for their efforts.

Author Response

In response to Reviewer 1

The authors thank the reviewer for the suggestions, and have answered the query to the best of our ability, namely:

It would be interesting to know the uses of the horses, although I acknowledge that this data may not be available. For example, 44% of horses were from breeds which are commonly used for high intensity exercise (quarterhorse, thoroughbred standardbred). Where these horses undergoing race training, in which case they would probably be stabled, or were they retired? What percentage of the other horses were used for athletic pursuits? As stated above this is not essential, and the data may not be available, but it could be of value to discuss these details if possible.

Unfortunately, as the reviewer notes, the information is largely unavailable.  However, we have, in lines 274-437 expanded on demographic data,

Reviewer 2 Report

Comments and Suggestions for Authors

The manuscript contains valuable information for practitioners especially related to owner observation of clinical signs of EA and opportunities for owner education related to nasal flaring and increased respiratory effort for horses with sEA. Most of my comments are related to presentation of the data in tables and figures to aid understanding and underscore important findings related to clinical presentation, lung function testing, BAL cytology and owner perception over time. 

Specific comments:

Abstract: 

Line 26: Consider including the number of cases of mEA and sEA.

Line 34: histamine challenge improved with treatment or time. 

Materials and methods: 

Line 171: In figure, please define what sufficient documentation includes and consider adding a box for mEA and sEA, the diagnostic criteria for each and the number of patients that had each diagnostic test performed. Consider splitting the box for patients with only 1 visit and patients with insufficient documentation into separate boxes. 

Results:

Please consider splitting the results into mEA and sEA and presenting the demographics (age median and range, sex, and breed) for each condition separately. Was either condition more common in males or females?

Adding a Table with variables like age, breed, sex, use, # of visits, and frequency of PE findings, rads, BAL, lung function testing, specific treatments would be helpful to visualize the vast amount of data reported here. Separating these into columns for mSE and sEA would also be helpful.

For the pie chart, consider presenting the number and percentage. Consider separating mEA and sEA cases. 

Including a table/figure to show changes in clinical presentation, lung function testing, BAL cytology and owner/vet perception over time for mSE and sEA would also be very helpful to visualize the data longitudinally. What number (%) got better, got worse and stayed the same for each metric?

It would be helpful to see a figure of the number of initial and recheck cases of mEA and sEA per study year. Is there a pattern of increasing or decreasing cases over the study period? If so, what are the possible reasons? Was there any seasonality in the initial exams or recheck exams?

In Table 1, can you further explain the directionality of the testing. Perhaps indicating which variables were more frequently observed by owners in table 1 would be useful.

Line 512: Have the authors considered a treatment bias as a reason for treating with steroids? Were any of the horses treated with environmental modifications only?

Discussion:

Line 515: In this study were horses with sEA older than horses with mEA?

Consider addressing the potential reasons for patients with only one visit in the discussion. Is it possible that the patients with > 1 visit came back because they were less likely to have fully recovered or are there other potential reasons? 

Author Response

The authors thank the reviewer for the suggestions, and have answered the query to the best of our ability, please see attachment.  

Reviewer 3 Report

Comments and Suggestions for Authors

This report addresses Equine Asthma Syndrome through a longitudinal analysis. The topic is interesting in equine medicine as equine asthma is a common disease in these patients.

The length and structure are correct, and in general terms, it is effortless to understand. However, there is some aspects that should be clarify before publishing it.

Tittle: it is a little bit confuse because seems that all the tests were performed in all the patients. Maybe would be better entitled the article as: a longitudinal analysis of equine asthma presentation using lung function testing or BAL cytology analysis and comparing with the owner perception. There was not a clear statiscal analysis between the different treatments and the outcome. 

Abstract

Line 217: the reference is not cited and not appeared in the references section (I guess is the ACVIM consensus from 2007)

Materials and methods

Line 164: the authors explain that not all the tests were done in all the horses. However, in line 167 they say that all the tests were done in all patients. This sentence should be changed to make it clearer.

Clinical examination scoring. Could you mind incorporate as supplementary file the questions and criterium that you used? I believe that it is very difficult perform this kind of score in a retrospective analysis

Line 256. Animals with samples obtained through tracheal aspirate should not be included because the cytology differences and the possibility to contaminate de sample with blood making the diagnoses of Pulmonary hemorrhage incorrect.

Results

Line 334: Only 1 horse had fever or only 1 horse was recorded with fever? If you have only data of temperature from 1 horse better remove the parameter from the study.

Finally, there are not tables and graphs illustrating the results and the data. It could be improve substantially the article including some of them.

Author Response

(The authors gave the same response as above.)

Reviewer 4 Report

Comments and Suggestions for Authors

The manuscript submitted for review is a retrospective presenting longitudinal data on horses with EA presented to Tufts Cummings School over a 24 year period. The author stated goals for the project were: “(1) to quantify and describe the types and range of clinical diseases that fall under the umbrella term of EA in horses in this geographic area, (2) to consider owners’ perceptions of their horse’s clinical signs and evaluate owner complaint scoring as a diagnostic tool, and (3) to document and assess the diagnostic methods used at Tufts Cummings School and assess the response of horses with EA to treatment and/or time.” The manuscript is well-written, informative and provides interesting results. The limitations are discussed appropriately. The authors have done an excellent job presenting a large amount of information in a very logical and succinct manner. I have only one potential request for additional data and a few minor suggestions that might improve readability and clarity for the first time reader.

Line 196 – “Complaints were determined by examining the record using the written history and the communications log.” Reading this description, it is unclear to me whether there was a systematic approach to gathering information on the criteria in the CS, or it was happenstance that that information was part of the patient history. Would it be possible, for example, for there to be lack of information on nasal discharge because the interviewer forgot to ask, the client forgot to mention it and there was none present on PE?

When explaining the agreement scores, it might be helpful to explain the number associated with best agreement vs. poor agreement. I know this can be reasoned out with the information provided, but a brief explanation of the meaning of a positive, neutral or negative score will allow the reader to understand the scores more readily.

“Respiratory effort” is referred to in the clinical examination scoring. In the discussion, this is further explained as abdominal effort. It would be helpful to explain what respiratory effort means to the evaluating clinician and to owners in the methods section, or that this parameter was open to interpretation.

I’m a bit confused by Table 1. Based on ACVIM consensus definitions, mEA horses do not have signs of respiratory effort or nostril flare at rest. I am assuming any events of these noted on mEA horse exams were noted during the exercise portion of the exam? It might be useful to remind readers of this when presenting these results? Follow up – this makes more sense after reading your discussion, but I wonder if you can get a preview of these discussion points into the results section so that some of the reader’s confusion is resolved earlier in the manuscript.

Line 452 – “…owners were actually more astute observers than the veterinarians…” I would respectfully submit that owners, who see their horses more frequently (noted in line 455-456) than the vet, have the benefit of more observations which creates more opportunities for noting cough. In my mind this is the advantage of more data creating greater accuracy, rather than astuteness.

Did PE include checking for inducible cough by rubbing the tracheal cartilages? If not, this technique might be commented on in the discussion lines 489-491? I find it to be very consistent in inducing cough in horses with sEA, but only variable with mEA.

Line 550-551: Suggest changing the sentence about owner vs veterinary perception to instead focus on owners having more observation data on their horses, and therefore having data that is more accurate, as an advantage that can be leveraged in “team-based” veterinary healthcare.

I noticed that there isn’t a table of BAL cytology data. As someone who sees EA in a different geographic region, I would be very interested to learn more about the BAL cytology in this population of horses. I would be particularly interested in the percentages of horses with “purely” mastocytic or eosinophilic inflammation vs. mixed or neutrophilic, and whether these horses were mEA or sEA. It is also interesting to know just how high those percentages go. For example, our record mastocytic mEA is 17%. But is that high, or just high for our region? I would be interested in whether or not the authors considered including, or would be willing to include, more of this data in this manuscript?

Author Response

(The authors gave the same response as above.)

Round 2

Reviewer 2 Report

Comments and Suggestions for Authors

The reviewer thanks the authors for their consideration of the previous comments.

Specific comments:

Line 43: Can you explain more clearly how tests of airway hyperreactivity are more successful than baseline lung function testing and BAL cytology in detecting changes in mEA? Does this mean that in the absence of the ability to perform tests of airway hyperreactivity, relying on owner perception is more reasonable than relying on repeat BALF cytology for follow up examinations? 

Line 355: If information on sex is not available in the medical record, please note this and discuss as a limitation especially in light of what is known regarding sex differences in asthma in humans, e.g. PMID: 34789462

Line 363: If ages are not normally distributed, median and range may be more appropriate than mean.

I maintain that including a Table 1 would provide the reader with a clear picture of the data. See for example, PMID: 31229583. Additionally, providing all of the data in a supplemental file would be extremely useful for other researchers. Thank you for providing this.

I agree with the authors that the new Figure 4 does not show evidence of trends over time and could be relegated to a supplemental file with the information that the "peak" years coincide with periods of active recruitment. 

Line 468: Please provide the directionality of the results of repeat BALs for the 35 horses that had repeat BALs performed. 

Author Response

Dear Editors and reviewers, we are grateful for your critique, and  have revised our manuscript accordingly.  Our letter describing our revisions point-by-point is attached.  Best regards, Melissa Mazan

Reviewer 3 Report

Comments and Suggestions for Authors

I would like to express my gratitude for the effort put into the corrections of this article. The work has improved substantially.

Author Response

Dear Editors and Reviewers, 

We are grateful for your critique, and have revised our manuscript accordingly, including Table 1.  Yours sincerely, Melissa Mazan
